# A New Route of Valorization of Petrochemical Wastewater: Recovery of 1,3,5-Tris (4-tert-butyl-3-hydroxy-2,6-dimethyl benzyl)–1,3,5-triazine-2,4,6-(1H,3H,5H)-trione (Cyanox 1790) and Its Subsequent Application in a PP Matrix to Improve Its Thermal Stability

**DOI:** 10.3390/molecules28052003

**Published:** 2023-02-21

**Authors:** Joaquín Hernández-Fernández, Rodrigo Ortega-Toro, Juan López-Martinez

**Affiliations:** 1Chemistry Program, Department of Natural and Exact Sciences, San Pablo Campus, University of Cartagena, Cartagena 130015, Colombia; 2Chemical Engineering Program, School of Engineering, Universidad Tecnológica de Bolivar, Parque Industrial y Tecnológico Carlos Vélez Pombo, Km 1 Vía Turbaco, Turbaco 130001, Colombia; 3Department of Natural and Exact Science, Universidad de la Costa, Barranquilla 30300, Colombia; 4Food Packaging and Shelf-Life Research Group (FP&SL), Food Engineering Department, Universidad de Cartagena, Cartagena de Indias 130015, Colombia; 5Institute of Materials Technology (ITM), Universitat Politecnica de Valencia (UPV), Plaza Ferrandiz and Carbonell s/n, 03801 Alcoy, Spain

**Keywords:** phenolic antioxidant, extraction, Cyanox 1790, circularity, recovery

## Abstract

The various chemicals in industrial wastewater can be beneficial for improving its circularity. If extraction methods are used to capture valuable components from the wastewater and then recirculate them throughout the process, the potential of the wastewater can be fully exploited. In this study, wastewater produced after the polypropylene deodorization process was evaluated. These waters remove the remains of the additives used to create the resin. With this recovery, contamination of the water bodies is avoided, and the polymer production process becomes more circular. The phenolic component was recovered by solid-phase extraction and HPLC, with a recovery rate of over 95%. FTIR and DSC were used to evaluate the purity of the extracted compound. After the phenolic compound was applied to the resin and its thermal stability was analyzed via TGA, the compound’s efficacy was finally determined. The results showed that the recovered additive improves the thermal qualities of the material.

## 1. Introduction

Pollution generated by petrochemical plants is of paramount concern today due to its adverse effects on the environment and human health [1,2,3,4]. The petrochemical industry is responsible for the emission of a wide variety of pollutants, including sulfur dioxide, polycyclic aromatic hydrocarbons (PAHs), and volatile organic compounds (VOCs), which contribute to climate change, to air pollution, and especially to the detriment of human health via the presence of carcinogens already determined, such as 1,3-butadiene and benzene, among others [1,2,3,5,6]. Water pollution is another problem commonly associated with petrochemical plants due to the accidental or intentional release of toxic chemicals into the water during production. Soil contamination can also be a problem in areas near petrochemical plants due to toxic residues and the possible infiltration of chemicals into the soil [7,8,9,10,11]. Pollution generated by petrochemical plants is a serious problem that requires effective measures to minimize its adverse effects on the environment and human health. Treating wastewater from petrochemical plants is of great importance due to the various chemical and toxic contaminants in this type of water [7,9,10,11,12]. Wastewater from petrochemical plants can contain hydrocarbons, heavy metals, acids, and other hazardous chemicals that can adversely affect the environment and human health if not treated properly [6,13,14,15,16,17]. Therefore, it is essential to implement effective treatments to remove these contaminants and ensure that wastewater meets the set quality standards before its discharge into the environment. COD (Chemical Oxygen Demand) and BOD (Biochemical Oxygen Demand) are critical parameters used to assess wastewater quality, especially in the context of petrochemical plants. Both parameters are important because they can provide valuable information on water’s organic load and biodegradation capacity, which can help determine the efficiency of treatment processes and assess the impact of wastewater on the environment [15,18,19,20].

Wastewater from petrochemical plants can contain various chemical and toxic contaminants, including volatile organic compounds (VOCs), polyhydroxyalkanoates (PHAs), phenols, and minerals. VOCs are a group of volatile organic compounds that can occur in wastewater from petrochemical plants due to the presence of refining and petrochemical processes. VOCs can adversely affect human health and the environment if not treated properly. On the other hand, PHAs are a group of biodegradable polymers produced from the biotransformation of organic matter in water and can be difficult to remove via conventional treatment processes [5,6,16,21,22,23,24]. Phenols can also be present in wastewater from petrochemical plants due to the presence of chemical processes and can have toxic effects on the environment and human health. Phenols are chemical compounds that are widely used in the petrochemical industry due to their unique properties and added value in producing various products [25,26,27,28]. Synthetic phenolic antioxidants (SPAs) are artificially manufactured chemical compounds with antioxidant properties similar to natural phenolic antioxidants (NPAs). SPAs can be used as additives in polymer manufacturing because they protect (inhibit) polymers from oxidative degradation [26,27,28,29,30,31]. SPAs can delay or prevent the oxidative degradation of polymers by neutralizing free radicals and other oxidizing agents [26,27,28,31]. SPAs are one of the families of emerging organic pollutants classified as anthropogenic, given their origin. Given the variety of uses and the breadth of their production, SPAs have been found in various environmental matrices such as marine sediments, river waters, and dust, among others. It has been shown that there is a migration of SPAs into the water from polymers. SPAs are also detected in urine and human (donated) sera at significant levels.

SPAs have been shown to generate toxic effects related to liver failure and damage to the endocrine system, and some could cause cancer. It is known that some of the SPAs, under certain conditions, can cause DNA strand breakage. SPAs are also of concern due to their bioaccumulation and high toxicity in aquatic environments [26,32,33,34,35,36,37,38,39,40,41]. 2,6-Di-tert-Butylphenol has been shown to pose a cancer risk [41]. Cyanox 1790 presents variants of 2,6-Di-tert-Butylphenol (the same variant three times) within its structure, as shown in Figure 1. Thus, given that this is a macromolecule, it could also represent a health risk with an increased multiplicity factor, so it is reasonable and indisputably necessary research that generates a knowledge base to establish the impact that this molecule can generate in the different aspects that concern us human beings (health, environment, economy), as is the present research. SPA removal from wastewater is essential due to its toxicity and resistance to biodegradation. Several techniques are available to remove SPAs from wastewater, including physical, chemical, and biological processes. Physical processes include adsorption and coagulation, which use materials such as activated carbon and polymers to remove phenols from water. Chemical processes include techniques such as chemical oxidation and neutralization, which rely on the use of chemical agents to oxidize or neutralize SPAs. Finally, biological processes include techniques such as biodegradation and biotransformation, which rely on the use of microorganisms to degrade SPAs into less toxic compounds [25,42,43]. Although several techniques are available to remove SPAs from wastewater, some may have deficiencies or limitations in their effectiveness [25,42,43].

SPA degradation is essential for removing these toxic compounds from wastewater, but some degradation techniques may need to be improved or improved in their effectiveness [44]. Biodegradation is a technique commonly used to degrade SPAs into less toxic compounds but can be limited by factors such as the availability of microorganisms and the presence of other contaminants that can inhibit microbial activity [42]. SPA recovery is a critical practice to leverage resources and minimize the environmental impact of the petrochemical industry. The recovery of phenols can contribute to conserving natural resources and reducing dependence on fossil fuels. SPA recovery can also reduce waste generation and decrease the pollutant load in wastewater. SPA recovery can also positively impact the economy, as it can provide an additional source of raw materials and reduce production costs [45,46,47]. Prior to the analysis of AOs in environmental samples in recent years, procedures such as liquid phase microextraction (LPME), solid phase extraction (SPE), and solid phase microextraction (SPME) were used in sample treatments [48,49,50]. According to the most recent quantitative and qualitative investigations of these AOs, which were performed with HPLC, the LOD is less than 1 ug L^−1^, and the relative standard deviation (RSD) is less than 10% [51,52,53,54] in matrices of low chemical complexity. The situation that we are in for this project is quite different from the aforementioned situation. Wei and colleagues (2011) quantified AOs in simulant C (10% ethanol) and simulant D (oil) using a C18 sorbent preconditioned with 5 mL of ACN and 5 mL of distilled water and obtained an LOD and an LOQ between 0.09 and 1.72 ug mL^−1^ and between 0.20 and 5.64 ug mL^−1^, respectively. With an RSD between 2.8 and 9.8%, their recovery ranged from 67.5 to 108.6%.

The selection of the instrumental technique to quantify these OA and its precision and analytical errors are of great importance for carrying out reliable measurements of these OA, given the protective function they have in the PP matrix to guarantee its thermal stability. Oxidative degradation is a process that occurs when polymers come into contact with oxygen and other oxidizing agents, such as light and heat, and degrade due to the formation of free radicals. Oxidative degradation can affect the service life of polymers and reduce their performance and properties [25,44]. Oxidative degradation can manifest itself in various ways, such as color change, strength deterioration, and the modification of mechanical and thermal properties. Oxidation is a process that can occur during the manufacture of polyolefins and can affect the quality and properties of polyolefins. Polymers can be sensitive to oxidation due to the presence of double bonds or carbonyl groups, which are vulnerable points to oxidation [29,30,55,56]. Due to its extremely low density, polypropylene (PP) is a polymer of petrochemical origin that is widely used in many sectors because it can be heated, cooled, and reheated without losing its composition. PP is characterized by its high mechanical strength, chemical resistance, thermal stability, and low production cost [26,27,57,58,59]. In addition, it is in high demand and is a product of great commercial interest, which drives production growth and can raise questions about potential environmental issues related to its manufacture and disposal. Phenolic antioxidants play an essential role in polypropylene’s increased resistance to oxidation [60]. Cyanox 1970, a phenolic antioxidant and non-bleaching stabilizer that works well on materials such as polyolefins and is recommended for polymer processing, is easily accessible [61,62,63,64]. Figure 1 shows the molecule in question from this research.

Unfortunately, many pollutants are produced in the PP production process with possible adverse effects on human health and the environment. These contaminants include synthetic phenols and volatile organic chemicals, which are added to PP as additives to enhance their qualities [27,55,65,66]. The existence of these materials in industrial waste supports the need to evaluate their processing efficiency and drives the creation of a more efficient and environmentally friendly PP production method. Deodorization is a step in manufacturing PP which consists of removing the substances that were not absorbed and that cause odor from the polymer. If these substances are recovered, they can be used in the PP extrusion process to give PP a new value, increase its manufacturing productivity, and reduce losses during its synthesis. This article proposes recovering the Cyanox 1970 additive and introducing it in the process to check if the recovered additive can improve the thermal qualities of PP.

In the present research, we intend to recover a high percentage (≥90%) of a high value additive (Cyanox 1790) for polymers and characterize its efficiency via competitive and reliable techniques to determine the feasibility of its reuse in the industry.

## 2. Analysis and Discussion

### 2.1. Identification, Quantification, Repeatability, Reproducibility, and Linearity Analysis of Multiple Cyanox 1790 Standards

This analysis is performed in a very detailed and rigorous way in some respects because, from the results obtained, it will be possible to determine if recovering this additive is workable.

#### 2.1.1. Multistandard Repeatability Analysis of Cyanox 1790 in CH_2_Cl_2_ and Multistandard per SPE in CH_2_CN

For the repeatability (precision) of the process, five tests were carried out using the HPLC-MS technique, with the same standard, on the same day, with the same analyst, depending on the relative standard deviation (RSD) and seven concentrations between 0 (white) and 5000 ppm. Accuracy within the day was validated if the average global values were less than 20%. A divergence of less than 15% from the expected value is suggested as an acceptable condition [66,67,68,69]. Table 1 shows the values and precision parameters calculated from the data obtained. These data were analyzed in an ANOVA and using Tukey’s method; it was found that all means were grouped in the same literal (A), which indicates that there are no significant differences in the means of the data, which allows us to infer with 95% certainty that the standard analysis is repeatable. Figure 2a shows a box plot to graphically observe the distribution of the repeatability data of the standard and the location of its means.

Table 2 shows the data obtained. We obtained RSDs much lower than 15%, where the highest was 3.6% (1500 ppm), errors less than 5.3%, and recoveries equal to or greater than 95% on all occasions, all of which were obtained by the same analyst on the same day. The highest yield obtained for these conditions was 98%. The ANOVA analysis performed for the SPE of Cyanox (standard), which is shown in Table 3, showed that the means did not have significant differences, and similarly to all the ANOVA performed so far, the grouping of the means occurred under the same bunk (A), which implies the significance of the difference between means. Figure 2c graphically complements the ANOVA set.

#### 2.1.2. Multistandard Reproducibility Analysis of Cyanox 1790 in CH_2_Cl_2_ and Multistandard by SPE in CH_2_CN

For the reproducibility of the process, five tests were performed using the same analytical technique with the same standard on different and consecutive days, with different analysts. For reproducibility, the same validation criteria apply for repeatability. As for repeatability and reproducibility, an ANOVA analysis was also performed (Table 3), with the same degree of confidence and under the same method (Tukey) shown in Table 1, and the same result was obtained as repeatability, that is, the grouping of all means in the same literal (A), which indicates that for the means of reproducibility, there is no significant difference either, that is, the process is reproducible. As for repeatability and reproducibility, Figure 2b shows the box plot for these data, which corroborate the ANOVA analysis.

For the reproducibility of the SPE, the same conditions are used. For reproducibility under the above conditions, we obtained a maximum RSD value of 5.2% (3000 ppm), errors less than 6.9%, and recoveries equal to or greater than 93%. In the case of errors, despite being higher than in the case of repeatability of the SPE, they are still below 15%, so it meets the established criteria. The ANOVA analysis (Table 3) performed for SPE of Cyanox (standard) under reproducibility conditions showed that the means did not have significant differences for the reasons explained by the previous ANOVA. Figure 2d shows the box plot for the reproducibility of the SPE in order to complete the information provided by the ANOVA visually.

#### 2.1.3. Repeatability Linearity and Reproducibility of Cyanox 1790 Multistandards

Figure 3a shows in the graph the theoretical concentrations established for SPE versus the concentrations obtained by the analyst. Each test has an independent linear regression that shows the same linear trend, in which a high degree of precision can be graphically demonstrated between the data. To obtain more specific information about (theory vs. reality), Figure 3b shows the theoretical values of the concentration of the SPE against the average of the data obtained per test, and here we find an (R2) of 0.99985, and a correlation coefficient of 0.99981, which allows us to establish first: that the model can satisfactorily predict the concentrations obtained experimentally, and second: that there is a relationship directly proportional between the theoretical and experimental values, representing the relevant fact that recovery values can be established with certainty, which is considered high, and this has significant economic and environmental implications.

As for repeatability, for the reproducibility of the SPE, the figure shows in Figure 3c the theoretical concentrations versus the experimental concentrations with their respective linear settings; here, it should be noted that there is a lower precision in the points corresponding to the concentration of 3000 ppm concerning the same concentration for the repeatability of the PSE. The graph in Figure 3d shows the theoretical values of concentration against the reproducibility averages, for which an (R2) of 0.99953 and a correlation coefficient of 0.99977 is obtained so that exactly the same conclusions can be reached as for the repeatability of the PES. All of the above was performed with the sole and exclusive purpose of guaranteeing the effectiveness and efficiency of the methodology to measure the concentration of Cyanox 1790 in actual samples.

### 2.2. Identification, Quantification, Repeatability, Reproducibility, and Linearity Analysis of Cyanox 1790 in Industrial Wastewater Samples

#### 2.2.1. Reproducibility Analysis Industrial Wastewater Samples

The actual samples were 40 in total, which were taken for 40 consecutive days and analyzed by five different analysts on the same day. For this sample, we obtained RSDs and the standard values below 15%, where the highest was 4.8%, and the recoveries were equal to or greater than 95.07%. The highest concentration value found in the samples occurred on day 26 and had a mean value of 4896 ppm, and the lowest value was found on day 38 with a mean value of 311.8 ppm, as shown in Table 4.

#### 2.2.2. Linearity and Distribution of Industrial Wastewater Sample Data

The first thing that should be appropriately noted of the sample is that during the forty days, totally different concentrations were found from one day to another, evidenced in the randomness of the values obtained, as shown in Figure 4a, in which the days are plotted against the average concentration of the sample obtained for each day. This high variability of concentrations is verified by the (R2) values and the correlation coefficient for the samples, which were 1.22 × 10^−4^ and 1.10 × 10^−2^, respectively. In Figure 4b, the samples are plotted against the values obtained for each analyst in each of the samples, and it is possible to observe that for each sample, the values of the analysts are not too far from each other, so graphically, the precision that they retain is evident. The values furthest from each other concerning the other samples correspond to samples 20, 21, 22, and 23. However, they remain within the acceptable range. In Figure 4c, we can see the average values recorded by analysts against the net recovery, all in ppm and with the values ordered from highest to lower. Here, we can appreciate the linear correspondence of the average values and the net recovery, which highlights that the methodology used for the recovery of our additive has a high recovery performance because the linear regression conducted for this dataset shows an (R2) of 0.99955 and a correlation coefficient of 0.99977. If the values obtained for the standards are taken as a reference, it is obtained that the square of the sample had an error of 0.002%, and the correlation coefficient had an error of 0%, because the same value was obtained.

If it is assumed that each sample contains 100% analyte, Figure 4d plots the samples against the percentage of recovery ordered from lowest to highest, such that a linear trend is seen. Once the difference in scales is more noticeable, separating the points concerning the linear regression—the trend and the minimum recovery value, which was 95.07%—is still essential. The ANOVA analysis (Table 3) performed for the concentration values of the samples obtained by the five analysts shows no significant differences between the means of the 40 samples under the same analysis conditions for the previous ANOVA. Figure 4e shows the box graph corresponding to the ANOVA analysis; here, you can see that there are outliers for four of the five analysts; apart from that, there is nothing.

### 2.3. Characterization of Recovered Dust and Comparison with a Pattern

#### 2.3.1. Thermogravimetry Analysis (TGA) of Pure Standard and Recovered Dust

In the thermal degradation of pure Cyanox 1790, the percentage by weight remained relatively stable at 100% until it reached 344 °C; from this point, the weight loss of the sample begins. The most significant percentage weight loss of this substance occurs between 344 and 445 °C from this temperature point. The degradation continues tenuously until the maximum temperature compared to the previous range. It is evident that the recovered Cyanox 1970 had exactly the same behavior as pure Cyanox in thermal degradation, so the analysis is the same for the recovered Cyanox, including the degradation range. The degradation starts temperature and attenuated degradation from 445 °C. All of the above is shown graphically in the TGAs in Figure 5. To corroborate the high degree of similarity between pure and recovered Cyanox, the pure and recovered DTGAs, respectively, were plotted in Figure 5 immediately below the TGAs, and here, the same maximum value is evidenced, precisely at the same temperature (396.59 °C, 0.138), for both derivatives. It indicates the tipping point for pure and recovered Cyanox.

This suggests that this additive is a fairly stable species in the surrounding environment (minimally reactive or reactive very specifically), because otherwise, changes would be expected to be at least perceptible in tests.

#### 2.3.2. Pure Standard DSC Analysis and Recovered Dust

Differential scanning calorimetry, similarly, to TGA and more similarly to DTGA, specifically in the graph aspect, shows the same peaks for pure and recovered Cyanox. The calorimetric curve shows that for both samples of Cyanox between 155 and 176 °C, there is heat release (peak up) from the sample, which is related to the decomposition of the sample, that is, the breaking of the bonds of the molecule, which release energy in the form of heat. For both Cyanox samples, the maximum heat flux released was 8.69 J/s, as shown in Figure 6.

#### 2.3.3. Pure Standard FTIR Analysis and Recovered Dust

Before and after reintegration of Cyanox into the PP to distinguish the spectra of the recovered substance, and after its addition to the PP to study the compatibility of the components in the composite matrix, both the recovered Cyanox 1790 and the pure Cyanox 1790 were subjected to Fourier transform infrared spectroscopy (FTIR). The spectrum obtained from the recovered Cyanox 1790 (Figure 7) [70] is remarkably comparable to the spectrum of the pure additive obtained from the literature, according to the first results of the FTIR before it joined the PP. Both spectra show the most intense peaks and the fingerprint of the spectrum in a comparable way. Given the noise of the signals acquired for this test and the low concentrations of the analyte, which are represented by lower absorbance values in the range of 2250–1900 cm^−1^, slight discrepancies in the spectra can be observed [71].

The resulting spectrum (Figure 7) shows a peak at approximately 1735 cm^−1^, which is indicative of the ester group (O=C) present in the structure of the Cyanox, and two bands between 1300 and 1050 cm^−1^, which correspond to the symmetrical and asymmetrical stretches of the Cyanox ester group. The intensity of one stretch is greater than that of the other. However, the signal at 3670 cm^−1^ shows that the group formed from phenols is present in Cyanox 1790. The usual band of the CH_3_ group is observed in the range of 2950 to 2970 cm^−1^. The chemical composition of Cyanox also reveals these categories. Another indicator of the presence of aromatic groups in the spectrum is a moderately strong absorption in the range of 1450–1500 cm^−1^, which is typical of the spectra of aromatic compounds. Because spectrophotometry is less desirable for this study because it is more desirable to measure the coupling and the difference between the peaks when the Cyanox enters a couple in the PP matrix, the analysis of the peak in the Cyanox spectrum allows us to confirm the presence of the recovered analyte, as well as its effectiveness. The molecular composition of Cyanox 1790 is shown in Figure 1 [71,72].

#### 2.3.4. Use of Recovered Dust and Application to Polypropylene Resins to Evaluate Their Efficiency

The integration of recovered Cyanox 1790 and pure Cyanox 1790 into virgin PP resin was evaluated by FTIR (Figure 8a). Figure 8b,c shows that the distinctive peak of the carbonyl functional group at 1735 cm^−1^ is absent from the IR spectra of virgin PP. According to Figure 7a,b, this functional group represents Cyanox 1790 on a regular basis. In the IR spectrum of virgin PP + pure Cyanox 1790 and virgin PP + Cyanox 1790 recovered, this carbonyl peak can be clearly recognized. Figure 7 shows more clearly the characteristic bands of the functional groups typical of the molecular structure (Figure 1) of Cyanox. The distinct phenol peaks, aromatic rings, and methyl groups are masked by the PP saturations in the infrared spectrum [3,27,28,72,73,74,75,76]. 

#### 2.3.5. OIT PP, (PP + Cyanox Pure), and (PP + Cyanox Recovered)

To determine the oxidation time, changes in the slope of the curve generated by the DSC with respect to the expected time and heat flow are taken into account. As shown in Figure 9, an endothermic peak was initially observed, but over time and with changes in the atmosphere, the slope of the curves changed, revealing a new exothermic behavior that is consistent with oxidation. It can be observed that, for unstabilized PP (Figure 9a), a significantly lower value of the induced oxidation time (OIT) is provided than when PP has a certain amount of additive. The oxidation time refers to the moment in which the displacement of this slope occurs. The value of OIT for the non-stabilized PP is 0.7 min, taking as reference the change in slope at 17.3 min, and the OIT for the PP with the added additive is 8 min, demonstrating how the presence of the additive slows down the oxidation processes of PP and improves its thermal stability due to its coupling in the polymer matrix. That mechanism reduces thermo-oxidation due to the presence of OH in its rings, which helps eliminate free radicals (Figure 9b,c).

## 3. Materials and Methods

### 3.1. Sampling

The samples were collected in a polypropylene manufacturing plant (Figure 10). Figure 10 shows three stages of the production process. Stage 1 corresponds to the receipt, purification, and storage of propylene. Tank 1 stores the untreated propylene. Column 2 is used for the elimination of traces of contaminants, and tank 3 stores the purified propylene. In stage 2, polymerization occurs, and propylene is transformed into polypropylene. This process happens at point 4, which corresponds to the polymerization reactor. Stage 3 corresponds to activation and pelletizing. Point 5 is a silo in which the virgin PP resin is received; at point 6, the virgin resin is mixed with the additives; at point 7, extrusion and pelletizing are carried out. Sampling is carried out at point 7. At this point, a flow of water comes out of the extruder. Water is used in the polymer extrusion process to lower the temperature and create a more uniform grain, as it is carried out at a high temperature [62].

Due to the large processing volume and fast processing speed, samples were obtained every 12 min, with a throughput of 5 samples. Approximately 30 tons of PP can be processed in 50 min. Given that the desorption process lasts four hours, samples of the desorption unit were obtained in duplicate every hour. This method investigates both the way phenols are present in the condensates of these two units and the impact of sampling time on phenol migration in condensates.

### 3.2. Extraction Solid Phases (SPE)

The sample was filtered via a polytetrafluoroethylene filter prior to the extraction (PTFE). Strata X-33 tubes (500 mg, 6 mL) were filled with 15 mL of the sample for the SPE at a flow rate of 1 mL/min. In total, 5 mL of methanol and 5 mL of distilled water were used to condition the filtrate earlier. MeOH: H_67_O 80:20 was used to wash the filtrate. The retentate was stripped with nitrogen at 5 psi after being eluted with 10 mL of acetonitrile (ACN). The extracted solution was dissolved in 1 mL of ACN [7] and then subjected to HPLC analysis. Until a concentrate volume of 500 mL was produced, the preconcentration procedure was repeated. By using a stream of N2, this 500 mL sample was preconcentrated and dried. The resulting solid was examined by FTIR, DSC, TGA, and HPLC-MS. For subsequent mixing of the recovered solid (Cyanox 1790) with virgin PP resin, it is kept in a humidity- and temperature-controlled cabinet.

### 3.3. HPLC-DAD/MS/MS (High-Performance Liquid Chromatography with Diode Array Detector and Mass Spectrometry)

For this analysis, an Agilent 1100 HPCL and a Micromass Quattro II triple quadrupole mass spectrometer were used. It is possible to acquire spectra using MS and MS/MS. As part of the system, a degasser (G1322A), a quaternary pump (G1311A), an automatic sampling system (G1313A), a column carrier (G1316A), a DAD detector (G1315B) with a chemical station, a Lichrosorb RP-18 column (4.6x200mmx5microns), 5 and 10 syringes, and a precision balance were used. A separate double pump system and self-sampler were also used for automatic injection into the MS with the Cyanox 1790 solution at ACN, which was used to establish chromatographic conditions. With the mixture of ACN and H2O solvents, which were mixed in various proportions, the following separation was carried out: 84 and 16 percent (1 min, 15 mL/min); 92 and 8 percent (2 min, 2 mL/min); 96 and 4 percent (3.5 min, 3.5 mL/min); and 100 and 0 percent (8 min, 3.5 mL/min). The temperature, irrigation volume, and wavelength of the column were adjusted to 50 °C. For the identification of the additive, the mass data of MS fragment and MS/MS ions are used [26,62,63].

### 3.4. The Recovered Additive Is Added to the PP Matrix

#### 3.4.1. Preparation of PP Sampling

The PP resin in question was produced from virgin resin devoid of additives. The recovered PP and Cyanox 1790 mixture samples were premixed by adding 0.1 wt% recovered Cyanox 1790 additive to PP powder using a standard Prodex Henschel 115JSS mixer at 800 rpm for 7 min at room temperature. The resin was then combined with the recovered Cyanox 1790. The samples were combined using a Welex-200 24.1 extruder, and melted extrusion was carried out at operating temperatures of 190, 195, 200, 210, and 220 °C on the extruder tracks. The mixtures underwent an additional transformation into films (300 mm diameter films with a thickness of about 100 mm) by compression molding in a hot press, the CARVER 3895. After the PP solid was granulated, 1000 mg/kg of recovered granules containing Cyanox 1790 were produced. Cyanox 1790 was diluted in ACN to solve with a concentration of 500 mg/L to manufacture the standard, and 2.5 mL of this solution was dissolved to a volume of 50 mL to obtain 25 mg/L.

#### 3.4.2. Fourier Transform Infrared Spectroscopy (FTIR)

A Nicolet 6700 infrared spectrometer was used for the FTIR study, with readings between 4000 and 600 cm^−1^ and a resolution of 2 cm^−1^ (reflection). The sample was pre-heated to 400 °C in preparation for this study to cause thermal deterioration and allow examination of changes in the polymer matrix [63,64].

#### 3.4.3. Differential Scanning Calorimeter (DSC)

The oxidation induction time (OIT) was calculated using DSC Q2000 V24.11 Build 124 equipment for calorimetric analysis. A 6.1 mg sample was used to obtain the results under atmospheric conditions of nitrogen and oxygen. The environment of the experiment was changed to examine how oxidation affects the volatility of the material. On the other hand, nitrogen provides a controlled and inert environment that allows us to examine how decomposition affects the sample. This procedure was carried out in various conditions, such as isothermal at 60 °C for 5 min, an atmosphere of 50 mL/min of nitrogen, and a temperature increase of 60 °C at 200 °C for 20 min. The ambient conditions of the analysis were then adjusted, and the sample was subjected for 30 min to an airflow of 50 mL/min under oxidation conditions at a temperature of 200 °C. A displacement of the slope of exothermic heat with oxidation flow was observed. The value of OIT corresponding to the time when the slope changes can be calculated using this transformation [63].

#### 3.4.4. Analysis by Thermogravimetry (TGA)

This analysis was performed using a Perkin Elmer TGA7 thermobalance at temperatures between 30 and 700 °C with a nitrogen flux of 50 mL/min. The temperature in the TGA at which 5% of the mass is lost was used to calculate the initial degradation temperature, and the DTG curve was used to determine the maximum degradation temperature [48,64,65].

## 4. Conclusions

There is evidence that industrial wastewater can be a source of numerous substances that can harm the environment and people’s health, but there is also evidence that it can be a source of substances that, once recovered, are very helpful in the industrial sector. Recovering these components, therefore, reduces the environmental effect and improves the circular economy. This study demonstrates how recovering phenolic compounds (Cyanox 1790), which are used as additives in polypropylene manufacturing, can yield high purity recoveries of over 98%. The thermal and thermo-oxidative stability of PP was significantly increased by the recovery and integration of Cyanox 1790 in the PP matrix. Without thermal stabilizers, PP would disintegrate throughout the extrusion process, preventing it from having any significant ultimate applications. Therefore, the recovery of this Cyanox 1790 at these high purity levels exemplifies a critical technology to be applied in the industrial sector and encourages sustainable raw material sources.

## Figures and Tables

**Figure 1 molecules-28-02003-f001:**
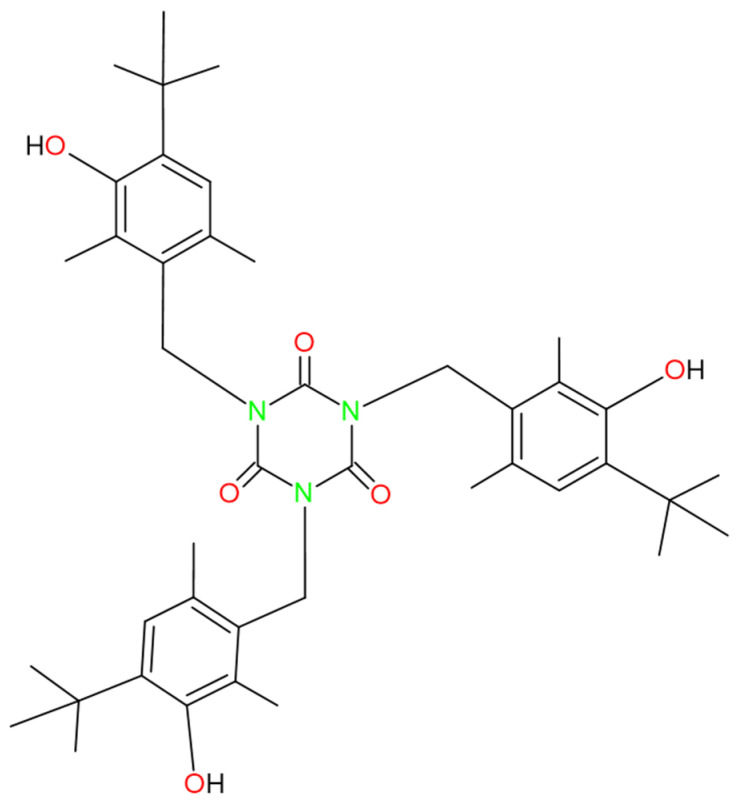
Chemical structure of Cyanox 1790.

**Figure 2 molecules-28-02003-f002:**
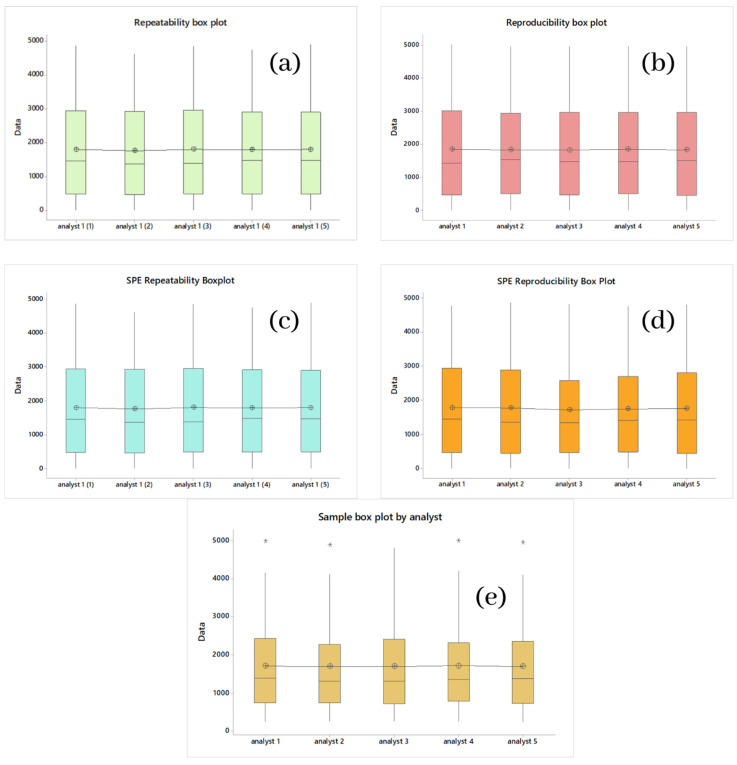
(**a**) Box Chart Repeatability of Banners; (**b**) Box Graph Reproducibility of Banners; (**c**) Box Graph Repeatability of Recovery of Banners; (**d**) Box Graph Reproducibility of Recovery of Banners; (**e**) Graph of Box Reproducibility of Samples. *, *p* ≤ 0.05.

**Figure 3 molecules-28-02003-f003:**
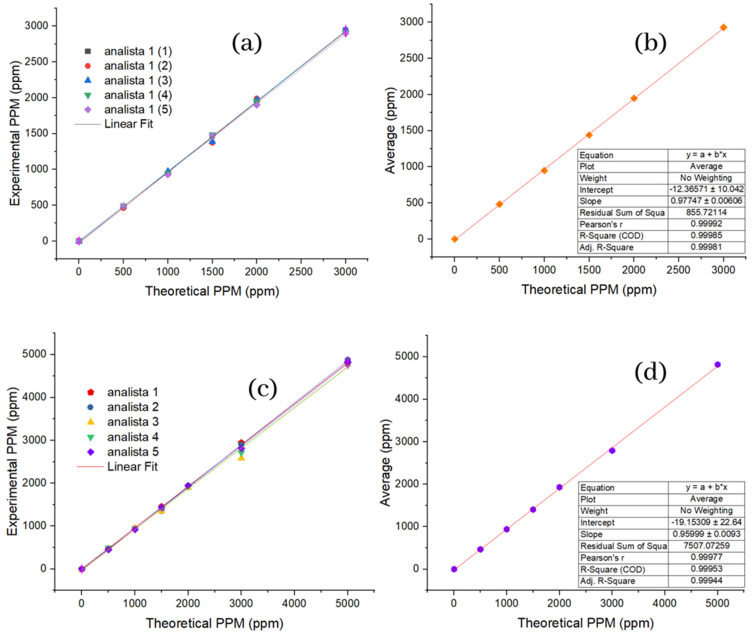
Linearity of interday and intraday measurements. (**a**) Repeatability for standard; (**b**) mean repeatability for standard; (**c**) reproducibility for standard; (**d**) mean reproducibility for standard.

**Figure 4 molecules-28-02003-f004:**
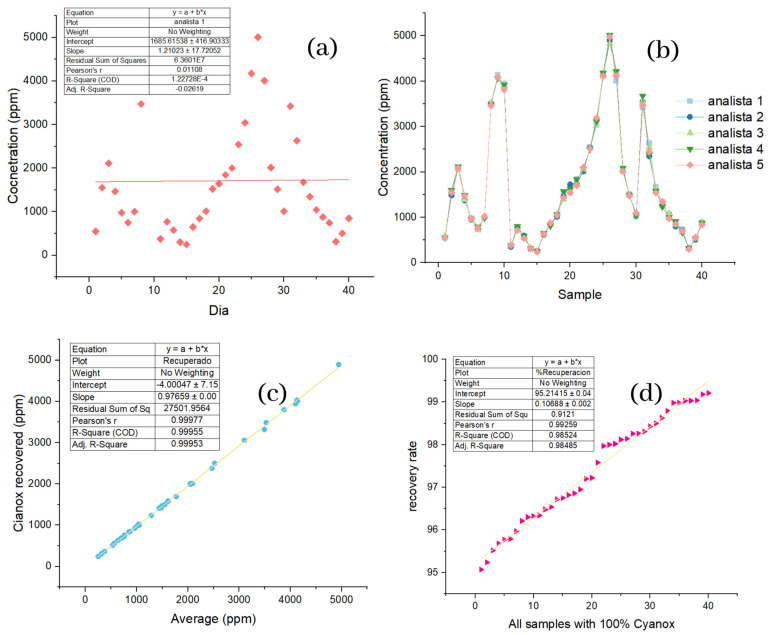
(**a**) Distribution of concentration data per day for samples; (**b**) repeatability of the data by analysts per sample; (**c**) analyte recovery based on mean sample concentrations; (**d**) percentage recovery of analyte based on 100% analyte per sample.

**Figure 5 molecules-28-02003-f005:**
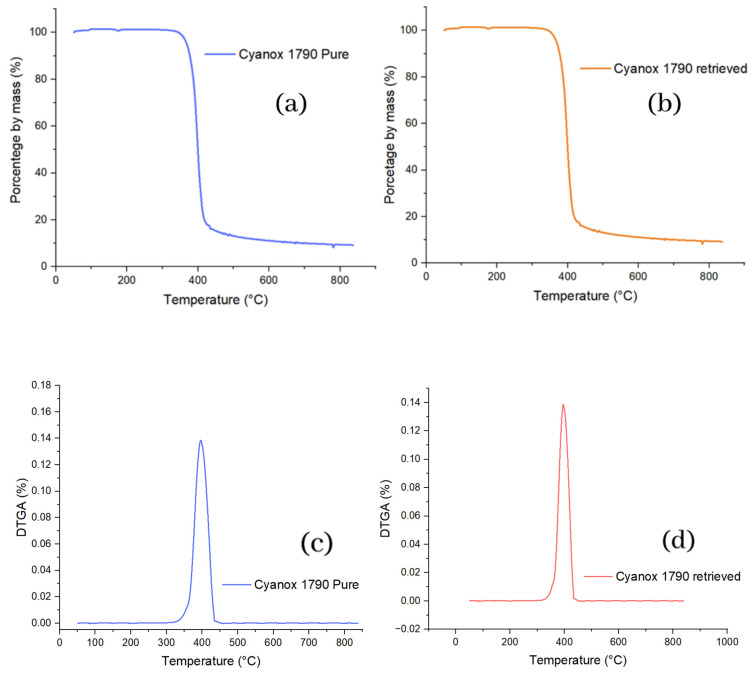
(**a**) TGA graphics of pure and (**b**) recovered Cyanox 1790 and (**c**) DTGA Cyanox 1790 pure and (**d**) recovered.

**Figure 6 molecules-28-02003-f006:**
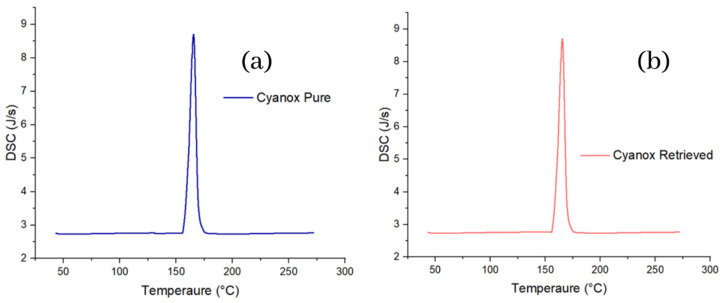
DSC charts for pure (**a**) and recovered (**b**) Cyanox 1790.

**Figure 7 molecules-28-02003-f007:**
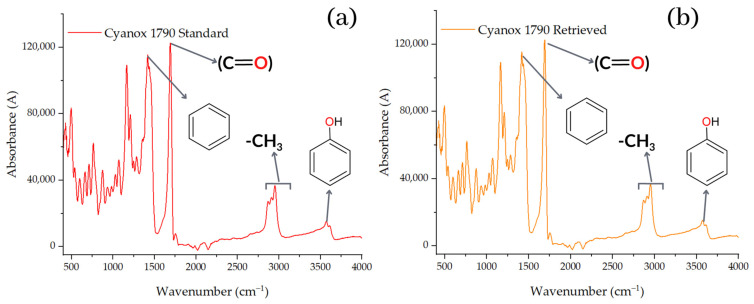
FTIR charts for pure (**a**) and recovered (**b**) Cyanox 1790.

**Figure 8 molecules-28-02003-f008:**
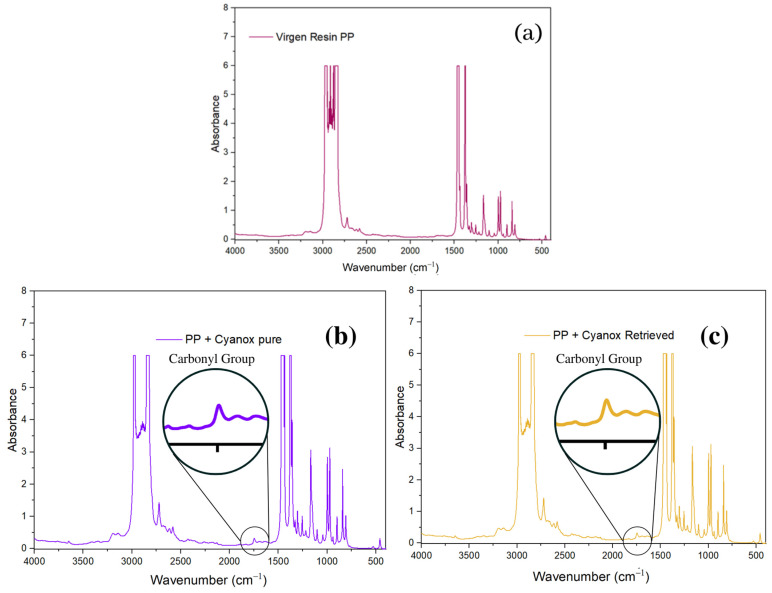
(**a**) Virgin PP infrared spectrum, (**b**) pure PP + Cyanox 1790 infrared spectrum, (**c**) recovered PP + Cyanox 1790 infrared spectrum.

**Figure 9 molecules-28-02003-f009:**
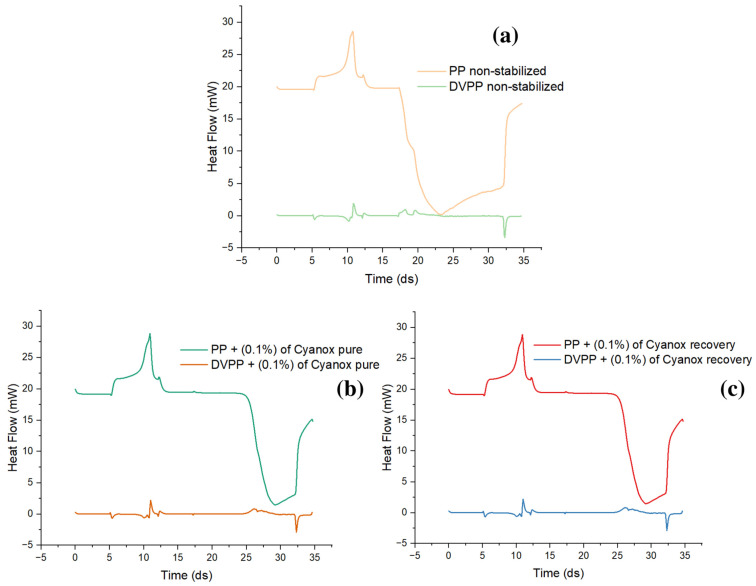
(**a**) Unstabilized PP oxidation induction time; (**b**) PP oxidation induction time with pure Cyanox 1790; (**c**) PP oxidation induction time with recovered Cyanox 1790.

**Figure 10 molecules-28-02003-f010:**
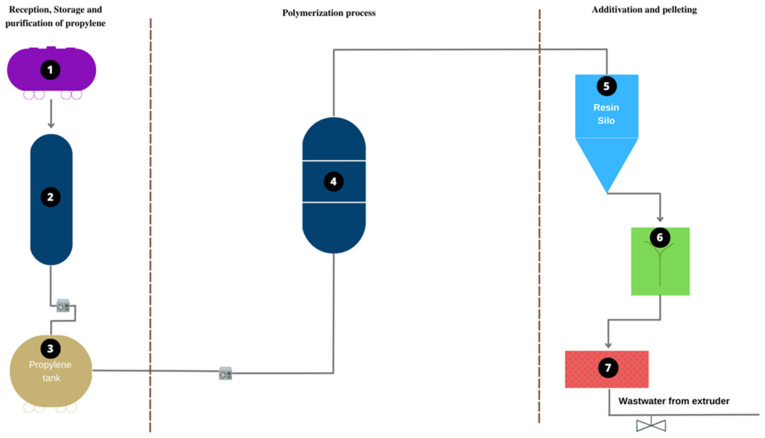
Flowchart of the plant from which the samples were taken.

**Table 1 molecules-28-02003-t001:** Tabulation of data for repeatability and reproducibility of standards in HPLC-MS.

Cyanox Calibration Curve with Dichloromethane on the HPLC-MS										
**Intraday Test (Same Day)**										
**Theoretical**	**STDA**	**Analyst 1**	**Analyst 1**	**Analyst 1**	**Analyst 1**	**Analyst 1**	**Average**	**Deviation**	**RSD**	**Error**
0	1	0.0	0.0	0.0	0.0	0.0	0.0	0.0	0.0	0
500	2	495.0	490.0	492.0	501.0	510.0	497.6	8.1	1.6	0.5
1000	3	942.0	972.0	981.0	1010.0	967.0	974.4	24.6	2.5	2.6
1500	4	1450.0	1514.0	1487.0	1491.0	1511.0	1490.6	25.6	1.7	0.6
2000	5	2012.0	1963.0	1976.0	1988.0	2013.0	1990.4	22.0	1.1	0.5
3000	6	2975.0	2989.0	3016.0	2991.0	2985.0	2991.2	15.2	0.5	0.3
5000	7	5021.0	4997.0	4987.0	5011.0	4991.0	5001.4	14.2	0.3	0.0
**Interday Test (Different Days)**										
**Theoretical**	**STDA**	**Analyst 1**	**Analyst 2**	**Analyst 3**	**Analyst 4**	**Analyst 5**	**Average**	**Deviation**	**RSD**	**Error**
0	1	0.0	0.0	0.0	0.0	0.0	0.0	0.0	0.0	0
500	2	475.0	501.0	476.0	501.0	453.0	481.2	20.3	4.2	3.8
1000	3	1015.0	947.0	958.0	1024.0	974.0	983.6	34.3	3.5	1.6
1500	4	1435.0	1542.0	1485.0	1476.0	1511.0	1489.8	40.0	2.7	0.7
2000	5	2024.0	1945.0	1975.0	2015.0	1986.0	1989.0	31.8	1.6	0.6
3000	6	3021.0	2946.0	2978.0	2976.0	2976.0	2979.4	26.8	0.9	0.7
5000	7	5032.0	4978.0	4967.0	4987.0	4969.0	4986.6	26.6	0.5	0.3

**Table 2 molecules-28-02003-t002:** Tabulation of data for repeatability and reproducibility of standards in the SPE.

SPE with Acetonitrile (HPLC-MS Results)											
Intraday Test (Same Day)											
Theoretical	STDA	Analyst 1	Analyst 1	Analyst 1	Analyst 1	Analyst 1	Average	Deviation	RSD	Error	%Recovery
0	1	0.0	0.0	0.0	0.0	0.0	0.0	0.0	0.0	0	0
500	2	478.0	462.0	486.0	498.0	490.0	482.8	13.7	2.8	3.4	97
1000	3	942.0	946.0	976.0	948.0	927.0	947.8	17.8	1.9	5.2	95
1500	4	1465.0	1375.0	1389.0	1486.0	1475.0	1438.0	51.9	3.6	4.1	96
2000	5	1929.0	1988.0	1976.0	1946.0	1900.0	1947.8	35.5	1.8	2.6	97
3000	6	2941.0	2937.0	2955.0	2913.0	2900.0	2929.2	22.3	0.8	2.4	98
5000	7	4876.0	4642.0	4863.0	4772.0	4912.0	4813.0	108.6	2.3	3.7	96
**Interday Test (Different Days)**											
**Theoretical**	**STDA**	**Analyst 1**	**Analyst 2**	**Analyst 3**	**Analyst 4**	**Analyst 5**	**Average**	**Deviation**	**RSD**	**Error**	**%Recovery**
0	1	0.0	0.0	0.0	0.0	0.0	0.0	0.0	0.0	0	0
500	2	468.0	452.0	472.0	488.0	451.0	466.2	15.4	3.3	6.8	93
1000	3	951.0	934.0	956.0	937.0	918.0	939.2	15.0	1.6	6.1	94
1500	4	1455.0	1356.0	1345.0	1425.0	1436.0	1403.4	49.6	3.5	6.4	94
2000	5	1941.0	1942.0	1900.0	1922.0	1942.0	1929.4	18.5	1.0	3.5	96
3000	6	2945.0	2901.0	2586.0	2712.0	2815.0	2791.8	145.5	5.2	6.9	93
5000	7	4786.0	4875.0	4825.0	4772.0	4821.0	4815.8	40.1	0.8	3.7	96

**Table 3 molecules-28-02003-t003:** ANOVA analysis for standards and samples using the Tukey method with 95% confidence.

Repeatability of Standards with CH_2_Cl_2_			
Factor	N	Average	Grouping
Analyst 1–3	7	1806	A
Analyst 1–1	7	1804	A
Analyst 1–5	7	1801	A
Analyst 1–4	7	1795	A
Analyst 1–2	7	1764	A
**Reproducibility of standards with CH_2_Cl_2_**			
Analyst 1	7	1857	A
Analyst 4	7	1854	A
Analyst 5	7	1838	A
Analyst 2	7	1837	A
Analyst 3	7	1834	A
**Repeatability of standards with CH_2_CN in SPE**			
Analyst 1–3	7	1806	A
Analyst 1–1	7	1804	A
Analyst 1–5	7	1801	A
Analyst 1–4	7	1795	A
Analyst 1–2	7	1764	A
**Reproducibility of standards with CH_2_CN in SPE**			
Analyst 1	7	1792	A
Analyst 4	7	1780	A
Analyst 5	7	1769	A
Analyst 2	7	1751	A
Analyst 3	7	1726	A
**Reproducibility of samples with CH_2_CN in SPE**			
Analyst 1	40	1718	A
Analyst 4	40	1710	A
Analyst 5	40	1699	A
Analyst 2	40	1696	A
Analyst 3	40	1695	A

**Table 4 molecules-28-02003-t004:** Data tabulation for sample reproducibility.

Analysis of Final Samples with SPE with Acetonitrile (HPLC-MS Results)											
Day	Sample	Analyst 1	Analyst 2	Analyst 3	Analyst 4	Analyst 5	Average	Deviation	RSD	Error	%Recovery
1	1	550	557	596	550	549	560.4	20.2	3.6	555	99.04
2	2	1552	1485	1561	1600	1548	1549.2	41.4	2.7	1500	96.82
3	3	2110	2078	2059	2115	2085	2089.4	23.2	1.1	2012	96.3
4	4	1465	1375	1389	1486	1475	1438	51.9	3.6	1413	98.26
5	5	978	948	956	967	975	964.8	12.7	1.3	924	95.77
6	6	748	733	715	791	765	750.4	29.3	3.9	718	95.68
7	7	1000	1012	1014	986	1021	1006.6	13.8	1.4	975	96.86
8	8	3474	3514	3520	3483	3465	3491.2	24.5	0.7	3319	95.07
9	9	4142	4085	4123	4065	4086	4100.2	31.4	0.8	3945	96.21
10	10	3945	3845	3865	3921	3811	3877.4	55	1.4	3800	98
11	11	375	348	396	373	382	374.8	17.5	4.7	361	96.32
12	12	768	777	745	800	715	761	32.4	4.3	755	99.21
13	13	575	596	542	585	536	566.8	26.5	4.7	561	98.98
14	14	300	312	324	317	313	313.2	8.8	2.8	310	98.98
15	15	245	263	260	246	235	249.8	11.6	4.6	241	96.48
16	16	645	615	635	650	631	635.2	13.6	2.1	630	99.18
17	17	842	865	849	812	876	848.8	24.5	2.9	834	98.26
18	18	1012	1015	1073	1062	1046	1041.6	27.4	2.6	1024	98.31
19	19	1522	1463	1400	1572	1429	1477.2	69.8	4.7	1415	95.79
20	20	1642	1725	1534	1629	1549	1615.8	77.4	4.8	1583	97.97
21	21	1842	1736	1742	1832	1700	1770.4	63	3.6	1691	95.52
22	22	2000	2015	2053	2075	2100	2048.6	41.4	2	2008	98.02
23	23	2541	2542	2530	2500	2510	2524.6	18.8	0.7	2500	99.03
24	24	3041	3145	3024	3108	3190	3101.6	69.7	2.2	3059	98.63
25	25	4174	4125	4108	4180	4110	4139.4	35	0.8	4013	96.95
26	26	5004	4900	4829	5010	4976	4943.8	77.7	1.6	4896	99.03
27	27	4005	4120	4200	4215	4119	4131.8	83.6	2	4032	97.58
28	28	2010	2042	2075	2090	2014	2046.2	35.8	1.7	1989	97.2
29	29	1512	1500	1498	1482	1475	1493.4	14.8	1	1470	98.43
30	30	1012	1042	1058	1043	1095	1050	30.2	2.9	1000	95.24
31	31	3421	3542	3541	3674	3465	3528.6	96.3	2.7	3486	98.79
32	32	2631	2345	2541	2384	2446	2469.4	116.8	4.7	2379	96.34
33	33	1674	1600	1631	1583	1543	1606.2	49.5	3.1	1576	98.12
34	34	1342	1263	1245	1234	1345	1285.8	53.7	4.2	1234	95.97
35	35	1042	1075	1093	1000	975	1037	49.5	4.8	1003	96.72
36	36	875	800	843	912	849	855.8	41.4	4.8	843	98.5
37	37	742	715	675	668	700	700	30.2	4.3	687	98.14
38	38	312	326	300	321	300	311.8	11.9	3.8	301	96.54
39	39	501	512	536	542	555	529.2	22.2	4.2	512	96.75
40	40	846	894	900	875	836	870.2	28.4	3.3	846	97.22

## Data Availability

Not applicable.

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
