# Peer review of "A New Route of Valorization of Petrochemical Wastewater: Recovery of 1,3,5-Tris (4-tert-butyl-3-hydroxy-2,6-dimethyl benzyl)–1,3,5-triazine-2,4,6-(1H,3H,5H)-trione (Cyanox 1790) and Its Subsequent Application in a PP Matrix to Improve Its Thermal Stability"

_molecules, 2023, doi:10.3390/molecules28052003_

Round 1

Reviewer 1 Report

The work is well planned and written, howeevr the follwing points to be considered in order to make improvements in the manuscript.

1. The flowchart (Figure 2) is not well explained and also its very comnfusing the flow chart. Please replace with a dschematic diagrm with more clear details and add the corresponding explanation.

2. Its not clear the FIgures 8 and 9, FTIR spectra. It is important to add or mention each peak its functional groupp and explain wiht proper reference , the fucntionl gorups and corresponding wave length. 

3. Finally there are numeros typo errors all over the manuscript, please revise and correct all the gramatical and english spelling mistakes.

Author Response

Dear

Thank you for reviewing this investigation. We have made all the necessary corrections and changes.

The work is well planned and written, howeevr the follwing points to be considered in order to make improvements in the manuscript.

1.The flowchart (Figure 2) is not well explained and also its very comnfusing the flow chart. Please replace with a dschematic diagrm with more clear details and add the corresponding explanation.

R/ Thank you for reviewing this paper. Now we have made the corrections.

  1. Its not clear the FIgures 8 and 9, FTIR spectra. It is important to add or mention each peak its functional groupp and explain wiht proper reference , the fucntionl gorups and corresponding wave length. 

R/ Thank you for reviewing this paper. Now we have made the corrections. Now we have changed Figure 8 and, for clarity, placed the functional groups of interest in the characteristic peaks of the spectra, which we have also explained. We have repeated the measurements, and some of the cyanox bands have been masked in the saturated bands of the polymer. We observe more clearly the characteristic peak of the carbonyl group.

  1. Finally there are numeros typo errors all over the manuscript, please revise and correct all the gramatical and english spelling mistakes.

R/ Thank you for reviewing this paper. Now we have made the corrections.

Kind Regards

joaquin

Reviewer 2 Report

The article deals with the valorization of a phenolic compound used in the processing of polypropylene. The topic is interesting and the results are good, however there are some typographic errors to fix and minor comments:

-line 8, there is an extra 1

-Line 122 L1 correct to L

-Line 126 and 127 mL-1, put -1 as superscript

- From line 129-130, I would connect better the sentences. The topic is different, the transition is harsh

-Line 162, adjust the punctuation

-Figure 2. Write stague in English

-Line 206 and line 210, 225 . H2O, N2 2 as subscript

-Line 253 cm-1, -1 as superscript

-line 345 Table 2. Convert the caption from spanish to english

-Line 357 I don't understand the symbol before relationship and line 479

-Table 3 CH2Cl2 2 as subscript and adjust the number of the rows that cover the text in the table

-Line 466 x10-4, -4 and -2 as superscript

-line 473. What does it mean from May to lower?

-Line 481 repetition, per sample for samples

-Line 486 DOnce, what is the meaning?

-Line 548, 550, 552, 554, 556, 567 -1 as superscript

-Line 553 CH3, 3 as subscript

-Line 566 figure 9a, there is an extra bracket and show instead of shows

-Figure 9. Is it possible to repeat the measurements or explain if the signal is in saturation to show better the peak of the additive

-Figure 10, caption PP instead of pp.

Author Response

Dear

Thank you for reviewing this investigation. We have made all the necessary corrections and changes.

The article deals with the valorization of a phenolic compound used in the processing of polypropylene. The topic is interesting and the results are good, however there are some typographic errors to fix and minor comments:

-line 8, there is an extra 1

R/ Thank you for reviewing this paper. Now we have made the corrections.

-Line 122 L1 correct to L

R/ Thank you for reviewing this paper. Now we have made the corrections.

-Line 126 and 127 mL-1, put -1 as superscript

R/ Thank you for reviewing this paper. Now we have made the corrections.

- From line 129-130, I would connect better the sentences. The topic is different, the transition is harsh

R/ Thank you for reviewing this paper. Now we have made the corrections.

-Line 162, adjust the punctuation

R/ Thank you for reviewing this paper. Now we have made the corrections.

-Figure 2. Write stague in English

R/ Thank you for reviewing this paper. Now we have made the corrections.

-Line 206 and line 210, 225 . H2O, N2 2 as subscript

R/ Thank you for reviewing this paper. Now we have made the corrections.

-Line 253 cm-1, -1 as superscript

R/ Thank you for reviewing this paper. Now we have made the corrections.

-line 345 Table 2. Convert the caption from spanish to English

R/ Thank you for reviewing this paper. Now we have made the corrections.

-Line 357 I don't understand the symbol before relationship and line 479

R/ Thank you for reviewing this paper. Now we have made the corrections.

-Table 3 CH2Cl2 2 as subscript and adjust the number of the rows that cover the text in the table

R/ Thank you for reviewing this paper. Now we have made the corrections.

-Line 466 x10-4, -4 and -2 as superscript

R/ Thank you for reviewing this paper. Now we have made the corrections.

-line 473. What does it mean from May to lower?

R/ Thank you for reviewing this paper. Now we have made the corrections. We change May to highest

-Line 481 repetition, per sample for samples

R/ Thank you for reviewing this paper. Now we have made the corrections.

-Line 486 DOnce, what is the meaning?

R/ Thank you for reviewing this paper. Now we have made the corrections. The correct word is Once.

-Line 548, 550, 552, 554, 556, 567 -1 as superscript

R/ Thank you for reviewing this paper. Now we have made the corrections.

-Line 553 CH3, 3 as subscript

R/ Thank you for reviewing this paper. Now we have made the corrections.

-Line 566 figure 9a, there is an extra bracket and show instead of shows

R/ Thank you for reviewing this paper. Now we have made the corrections.

-Figure 9. Is it possible to repeat the measurements or explain if the signal is in saturation to show better the peak of the additive

R/ Thank you for reviewing this paper. We have repeated the measurements, and some of the cyanox bands have been masked in the saturated bands of the polymer. We observe more clearly the characteristic peak of the carbonyl group.

-Figure 10, caption PP instead of pp.

R/ Thank you for reviewing this paper. Now we have made the corrections

Kind Regard

Joaquin

Round 2

Reviewer 2 Report

Adjust figure 2, receipt with the capital letter and storage instead of storag.

Author Response

Dear,

Thank you for your time in reviewing this research.

We have now corrected the errors in the Figure